# Construction of Cultivated Land Ecological Network Based on Supply and Demand of Ecosystem Services and MCR Model: A Case Study of Shandong Province, China

**Yifan Xu [1], Yuepeng Liu [1], Qian Sun [1,2] and Wei Qi [1,*]**

[1] College of Resource and Environment, Shandong Agricultural University, Taian 271018, China; xuyf@sdau.edu.cn (Y.X.); liuyp@sdau.edu.cn (Y.L.); applesq@sdau.edu.cn (Q.S.)

[2] College of Information Science and Engineering, Shandong Agricultural University, Taian 271018, China

[*] Correspondence: qiwei@sdau.edu.cn

**Abstract:** The research on the ecological protection of cultivated land has gradually become a focus and frontier of cultivated land protection. Constructing an ecological network of a cultivated land system is important to improve the effect of cultivated land ecological protection. In this study, the supply-demand ratio of five ecosystem services was calculated from 2000 to 2020 in Shandong Province, a major grain-producing area in China. The morphological spatial pattern analysis (MSPA) model was used to construct the minimum cumulative resistance (MCR) model. The conclusions of the study are as follows: (1) the areas with a higher supply and demand ratio of various ecosystem services of cultivated land in Shandong Province are distributed in southern and southwestern areas. (2) The ecological source of cultivated land in Shandong Province has decreased by about 7000 km$^2$ from 2000 to 2020, and the fragmentation trend is obvious. (3) The total length of the ecological corridors is 798.5 km$^2$. The majority of these corridors are located in central and southern Shandong. The ideas and conclusions of this study are important for the formulation of ecological protection policies for regional cultivated land systems.

**Keywords:** ecosystem services; MCR model; InVEST model; morphological spatial pattern analysis (MSPA)

## 1. Introduction

Cultivated land is a valuable resource on which human beings depend for their survival and development and is an important insurance for national food security. China's unprecedented urbanization has resulted in a significant decrease in cultivated land area. The loss of cultivated land not only threatens China's food security, but also leads to ecosystem degradation [1]. From 1998 to 2008, the total loss of ecosystem service in the North China Plain caused by the reduction of cultivated land led to a loss of 34.66% of the total ecosystem service value [2]. Nearly 40 years have passed since 1986, when the State explicitly included the protection of cultivated land as one of the basic national policies of China. During the past 40 years, the theoretical exploration of cultivated land protection has not stopped. The focus on cultivated land protection has also shifted from quantity to quantity and quality, and in 2017, the State Council issued the Opinions on Strengthening Cultivated Land Protection and Improving the Balance of Occupation and Compensation, which formally proposed to build a new pattern of cultivated land protection that is based on the three aspects of quantity, quality, and ecology [3]. The utilization of cultivated land in China is shifting from the increasing grain production to improving the quality and function of cultivated land. With the rise of the concept of *ecosystem service* (ES), the research on the ecological protection of cultivated land has gradually become a hotspot and frontier of cultivated land protection. China's agricultural practices are currently characterized by the extensive use of chemical fertilizers and pesticides [4]. The huge amount of chemical

fertilizer and pesticide residues has seriously jeopardized the ecological effects of the cultivated land systems. Rough cultivation and unscientific management modes have also caused serious degradation to cultivated land in China. In order to ensure the sustainability of China's cultivated land system, and to implement the national strategy of "hiding grain in the land", the ecological management of cultivated land protection should be vigorously promoted [5].

The theoretical system and implementation path of the quantity and quality protection of cultivated land have been basically established [6]; however, the ecological protection of cultivated land is still in the stage of theoretical exploration. Song and others have analyzed the value of cultivated land resources in the Northeastern China dark soil from the perspective of the "three lives" space [7]. Zhao et al. advocate that the existing line of cultivated land protection should be shifted from single function to multi-function. By comprehensively analyzing the functions of cultivated land, the quality, quantity, and ecological protection of cultivated land should be raised to the protection of cultivated land with multiple functions [8]. Liu et al., on the other hand, utilized GIS spatial analysis methods and landscape pattern analysis to explore the characteristics of spatial patterns of cultivated land and to make corresponding suggestions for the protection and development of cultivated land [9]. Han et al. studied the landscape ecological security of cultivated land and its characteristics in the county, and explored the impacts of natural and socioeconomic factors on the landscape ecological security of cultivated land in the region [10]. Song and Deng investigated the impacts on ecosystem services of the transformation process from cultivated land to urban land in the North China Plain on the basis of the net primary productivity-based ecosystem services model (NESM) and the buffer zone comparison method impacts [11]. Huang calculated the ecosystem service value of cultivated land from 2000 to 2015 in order to explore the spatial and temporal change law of the cultivated land ecosystem service supply capacity in China [12]. Li argued that there have been many studies on multi-functional cultivated land assessment, but the expansion of the results of multi-functional cultivated land assessment is limited, and there are fewer studies on the categorization of cultivated land use and differentiated use management [13]. Sun argued that ecological products and services can be used as results or actions to incentivize cultivated land ecological protection. The principle requirements of ecological products are also summarized [14].

Numerous scholars have also researched the ecological protection of agricultural land. Kienast took the linkage of ecological consistency between the production system and the underlying ecosystems as a criterion for evaluating sustainable agriculture [15]. Gómez Sal and González García reviewed 101 studies that used quantitative methods to assess landscape or ecosystem multi-function, distilling how multi-function in agricultural landscapes is conceptualized, characterized, and quantified [16]. By exploring changes in the multi-functional aspects of ES, Frei et al. explored other social-ecological system resilience characteristics over time, concluding that the establishment of diverse, multi-functional agricultural landscapes is an important goal for overall system resilience and future food security [17]. Gaba. et al. found in their study that agroecology (ecological agriculture) requires an understanding of the relationship between biodiversity, functioning, and management, and takes into account the agricultural, ecological, and social linkages between them [18]. Hölting et al. integrated concepts such as agro-ecosystems across multiple spatial scales [19]. McGranahan reviewed the ecological rationale behind several concepts that are critical for reconciling food production and biodiversity conservation [20].

Ecosystem service is the bridge between research on sustainable land use and human well-being and is a central focus of sustainability research. Promoting land use planning through ecosystem service (ES) conservation is an important approach to maintain landscape sustainability. To achieve sustainability at the regional level, the provision of regional ecosystem services and the enhancement of the resilience of regional natural and social ecosystems should be strengthened. A clear understanding of the concepts related to ecosystem services and the existing research results can help the research on landscape

sustainability [2]. The InVEST model is widely used to assess ecosystem services from various perspectives. It provides different assessment modules, enabling the quantification, visualization, and analysis of ecosystem services. This helps in gaining a comprehensive understanding of the current state of the ecosystems, improving land use efficiency, and enhancing overall ecosystem services benefits. Researchers have utilized this model to study temporal and spatial changes in ecosystem service functions in different locations, including cities, watersheds, and ecological zones [19–21]. However, most existing studies have focused on the response of ecosystem services to historical land use changes. Further research is needed to understand the future changes in both land use and ecosystem services [22–25].

In recent years, morphological spatial pattern analysis (MSPA) has been widely applied to the study of pattern changes in landscape patterns such as forests and urban green spaces, but it has been less applied to cultivated land. The method is based on mathematical morphology and relies on morphological operations such as erosion and expansion to automatically segment the foreground elements of a raster binary image at the pixel scale into mutually exclusive types representing different sizes, shapes, and degrees of connectivity [26] including cores, islands, perforations, branches, bridges, roundabouts, and edges. The significance of the different morphological pattern types of MSPA in terms of the indications of morphological changes in cultivated land is determined according to their characteristics and meanings [27].

In this study, the ecosystem service capacity of the cultivated land system in Shandong Province was accurately described by calculating the supply and demand of five ecosystem services, and conducting spatial display and identifying spatio-temporal changes from 2000 to 2020. Combined with the utilization of the MSPA model, the core areas of the cultivated land ecosystem were divided using the morphological perspective analysis. Based on these, the MCR model was established to analyze and calculate the ecological security pattern of cultivated land in Shandong Province. The objectives of this research are: (1) exploring the supply and demand of ecosystem services, spatial and temporal distribution, and trends in cultivated systems in Shandong Province. (2) Determining the core service area of cultivated ecosystems in Shandong Province from the perspective of morphology. (3) Extracting ecological corridors to provide scientific basis for ecological protection of cultivated land in Shandong Province. The research results point out the direction for the ecological protection of cultivated land system in the main grain production areas of China and provide a scientific basis for the further establishment of relevant policies.

## 2. Materials and Methods

### 2.1. Study Area

Shandong Province (34°22.9′–38°24.01′ N, 11°7.5′–122°42.3′ E) is an important province on the eastern coast of China. Shandong is surrounded by the Yellow Sea and Bohai Sea to the east and north. Shandong is connected with Anhui Province, Jiangsu Province, Henan Province, and Hebei Province. According to the Third Land Use Survey of Shandong Province, the province's land area is 155,800 square kilometers, accounting for 1.54% of the country's total area. The plains are the main geomorphologic feature within Shandong Province, and their area accounts for about 67% of the total area of Shandong Province. The northwestern, western, and southwestern regions of Shandong Province all belong to the North China Plain. The landforms can be roughly divided into three parts: the mountainous and hilly area of central and southern Shandong, the lower mountainous and hilly area of eastern Shandong, and the plain area of northwestern Shandong. In general, the topography of Shandong Province is characterized as "high in the middle and low in the surroundings" [28]. The center of Shandong Province is the mountainous region dominated by the Taishan Mountain Range, and the highest point in Shandong is Yuhuangding, the highest peak in the Taishan Mountain region, with a height of 1545 m. To the east of Shandong Province are the hills of the Shandong Peninsula, which extend into the sea (Figure 1).

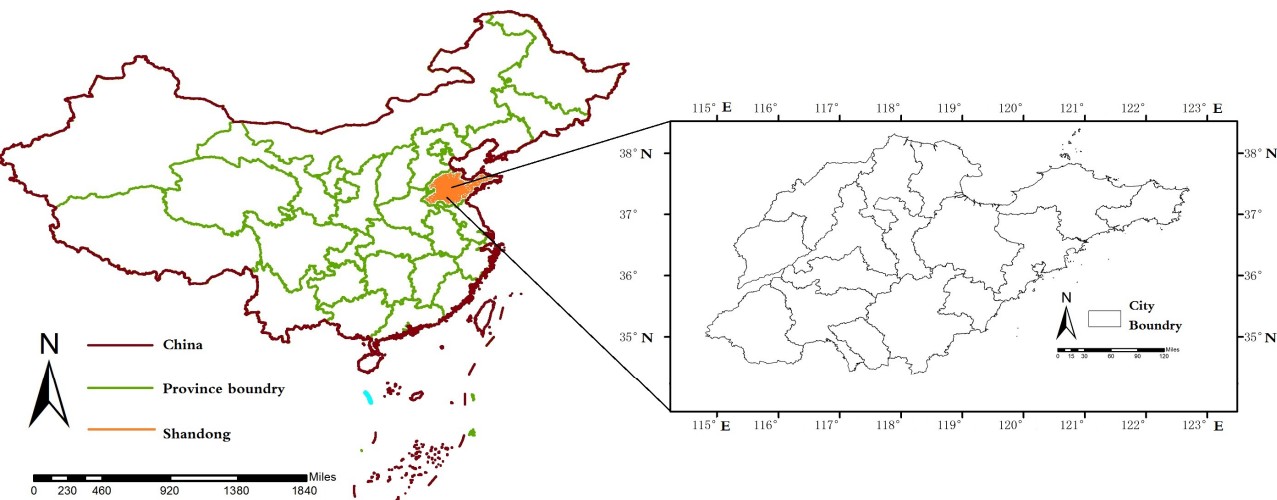

**Figure 1.** Map of the study area.

*2.2. Data Sources*

Land use data for Shandong Province for 2000–2015 were obtained from the ESA website. Land use data for Shandong Province for 2020 were obtained from the European Copernicus Meteorological Data Center. This data uses the same satellite data sources and image identification and processing as the ESA data and is in good agreement with the 2000–2015 data. DEM elevations were obtained from the Geospatial Data Cloud. Slope, slope direction, etc. were extracted from DEM by using ArcGIS 10.2.2 software.

The NDVI data of the year 2000/2005/2010/2015/2019 were obtained from the China Monthly Vegetation Index Spatial Distribution dataset from the Resource and Environment Science Data Center of the Chinese Academy of Sciences. The annual NPP was obtained from the Earth Observation Group, the National Center for Environmental Information, NCEI, under NOAA, the National Oceanic and Atmospheric Administration, USA. Temperature data were derived from the monthly value dataset of China's surface climate information. The annual solar radiation data are derived from the daily value dataset of basic elements of meteorological radiation in China. The precipitation data of Shandong Province from 2000 to 2020 are derived from the 1-km resolution month-by-month precipitation dataset of China by Peng of the National Tibetan Plateau Scientific Data Center [29–31].

Shandong Province 2000/2005/2010/2015/2020 GDP spatial distribution km grid data and 2000–2019 population spatial distribution data are from the Resource and Environment Science Data Center of the Chinese Academy of Sciences. Grain production in Shandong Province from 2000 to 2020 was obtained from the social and economic statistical yearbooks of counties (cities) in Shandong Province in the corresponding years (Table 1).

**Table 1.** Data sources and specifications.

| Data | Attribute | Spatial Resolution | Source |
|---|---|---|---|
| LUCC data | Year 2000/2005/2010/2015/2020 | 1 km | http://maps.elie.ucl.ac.be/CCI/viewer/ (accessed on 15 November 2023) https://cds.climate.copernicus.eu (accessed on 7 November 2023) |
| Climate | Average annual temperature Average annual precipitation | 1 km | https://cds.climate.copernicus.eu/ (accessed on 7 November 2023) |
| Terrain | STRM—DEM | 1 km | https://www.gscloud.cn/ (accessed on 1 November 2023) |

**Table 1.** *Cont.*

| Data | Attribute | Spatial Resolution | Source |
|---|---|---|---|
| Social and economic | GDP | 1 km | https://www.resdc.cn/ (accessed on 7 December 2021) |
| | Population | | https://www.resdc.cn/ (accessed on 23 November 2023) |
| Ecological | NDVI | 1 km | https://www.resdc.cn/ (accessed on 7 November 2023) |
| | NPP | 1 km | https://ngdc.noaa.gov/eog/index.html (accessed on 7 November 2023) |

*2.3. Methods*

2.3.1. The Calculation of Ecosystem Service Supply

1. Food production (FP)

Studies have shown that there is a significant positive correlation between the normalized vegetation index (NDVI) and the yield of agricultural products {Hou, 2021 #1}. Firstly, by consulting the yearbooks of each county and district of Shandong province, the grain production of each county in the five periods of 2000, 2005, 2010, 2015, and 2020 was counted. Then the Zonal Statistics function in ArcGIS was used to calculate the sum of NDVI on cultivated land plots in each county in each year. Divide the grain production by the NDVI of each county and then multiply the raster image by the NDVI raster image of each year with the raster calculator to get the map piece of grain production of each year [31]. The equation used is as follows:

$$FP_{supply} = \frac{P_i \times NDVI_1}{NDVI_0} \tag{1}$$

2. Habitat quality (HQ)

As an ecosystem, cultivated land systems provide a place and a material basis for the survival and reproduction of different species. By analyzing regional land use, it is possible to quantify the degree of threat to biodiversity. In this study, the Habitat Quality module of the InVEST model was used for the assessment of habitat quality [25]. The formula is:

$$Q_{mi} = H_i \left[ 1 - \left( \frac{D_{mi}^z}{D_{mi}^z + K^z} \right) \right] \tag{2}$$

where $H_i$ is the adaptation level of habitat type i, $K$ is the half-saturation constant, $Q_{mi}$ is the habitat quality of grid m in habitat type i, and $D_{mi}^z$ is the stress level of grid m in habitat type *i*.

The level of habitat quality represents the level of habitat maintenance function. In the model setup, the habitat types were divided into four categories: sloping cultivated land, paddy field, plain cultivated land, and non-cultivated land, and the habitat sensitivity was set to 1.0, 0.9, 0.8, and 0, respectively. Among them, the slope cultivated land and plain cultivated land were demarcated by a slope of 5° [25].

3. Water yield (WY)

Water is mainly manifested in river flow, stagnating floods and replenishing depletion, and ensuring water quality. Cultivated land plays an important role in purifying water quality, intercepting precipitation and regulating regional water cycles. In this paper, we assume that the cultivated land has runoff. The water quantity decomposition modeling method is used to simulate the water conservation function {González-García, 2020 #36}, and its calculation formula is:

$$WY = P - ET \tag{3}$$

$$ET = \frac{P\left(1 + \omega\frac{PET}{P}\right)}{1 + \omega\frac{PET}{P} + \left(\frac{PET}{P}\right)^{-1}} \tag{4}$$

where WY is the water yield, ET is the annual evapotranspiration, P is the annual rainfall, PET is the annual potential evapotranspiration, PET_m is the monthly average potential evapotranspiration, and ω is the land cover influence coefficient, assigned to 0.5. Based on the previous analysis, the potential evapotranspiration PET_m was calculated using the Hargreaves model with the formula:

$$PET_m = 0.0023\frac{R_a}{\lambda}(T + 17.8)\sqrt{T_x - T_n} \tag{5}$$

where $PET_m$ is the monthly potential evapotranspiration, and $PET_m$ is equal to the annual potential evapotranspiration divided by 12. $R_a$ is the top-of-atmosphere radiation in MJ/m$^2$, $\lambda$ is the latent heat of vaporization of water, which is taken to be 2.45 MJ/kg, $T_x$ and $T_n$ are the monthly minimum and maximum air temperature, respectively, and $T$ is the monthly average air temperature.

4.    Carbon sequestration (CS)

Carbon sequestration is an important regulating service to control atmospheric carbon concentration. Previous studies have shown that the ability of vegetation to sequester carbon is closely related to net primary productivity (NPP). Li found that according to the photosynthesis equation, vegetation fixes 1.63 units of carbon per unit of NPP accumulated [32]. Therefore, the formula for the carbon sequestration and oxygen release capacity of cultivated land is as follows:

$$CS = NPP \times 1.63 \tag{6}$$

5.    Soil retention (SR)

Soil and water conservation is the role of ecosystems (e.g., forests, grasslands, etc.) in reducing soil erosion due to water erosion through their structures and processes, and the evaluation of this indicator identifies the current and future priority areas that will bear soil and water conservation functions. Soil and water conservation functions are mainly related to climate, soil, topography, and vegetation. The amount of soil and water conservation (the difference between the potential soil erosion amount and the actual soil erosion amount) is used as the evaluation index, and the modified soil erosion equation (RUSLE) is used for the calculation [32], and the specific formulas are as follows:

$$A_c = A_p - A_r = R \times K \times L \times S \times (1 - C) \times P \tag{7}$$

where $A_c$ is the amount of soil and water conservation (t/hm2-a); $A_p$ is the amount of potential soil erosion; $A_r$ is the amount of actual soil erosion; R is the rainfall erosion factor; K is the soil erodibility factor; L and S are the topographic factors, where L stands for the slope length factor, and S stands for the slope gradient factor, and the degree of undulation is used for the topographic factor in this study; C is the vegetation factor. P factor is 0.5 for cultivated land.

2.3.2. The Calculation of Ecosystem Service Demand

1. Food demand was obtained by consulting the yearbook of Shandong Province. The total demand for food production is equal to the population density multiplied by the average demand for food production and the supply-demand ratio is calculated as:

$$FP_{demand} = food\ consumption\ per\ capita \times population\ dentisity \tag{8}$$

$$ESDR = \frac{S_i - D_i}{(S_{max} + D_{max})/2} \tag{9}$$

2. The demand for habitat quality reference refers to the research results of Peng [29]. The demand for ecosystem services is expressed in terms of the proportion of built-up land, population density and economic density with the following formula [33]:

$$D = P_i \times \lg R_i \times \lg E_i \tag{10}$$

where D denotes the ecosystem service demand index, Pi denotes the proportion of construction land in region i, expressed as a percentage %, Ri denotes the population density in people/km$^2$, and Ei denotes the economic density in million yuan/km$^2$.

3. The demand for water yield is derived by multiplying the residential water consumption of Shandong Province by the population density in the past years.

$$W_d = P_i \times Y_i \tag{11}$$

where Wd represents the human demand for water, Pi represents the regional population density in year i, and Yi represents the per capita water use of residents in year *i*.

4. The carbon sequestration demand is calculated based on the per capita carbon emissions of Shandong Province in each year from 2000 to 2020, and the emission data of each type of energy and the data of the resident population in each year are obtained by checking the website of the China Bureau of Statistics. Based on the total energy consumption of Shandong Province from 2000 to 2020, the total carbon emissions of Shandong Province in each year are obtained by multiplying the standard coal factor by the carbon emission factor. Divided by the resident population of Shandong Province in each year, the per capita carbon emissions of Shandong Province can be obtained and then combined with population density, the carbon sequestration demand of Shandong Province can be obtained.

5. Soil and water conservation is the role of ecosystems (e.g., forests, grasslands, etc.) in reducing soil erosion due to water erosion through their structures and processes and is calculated using the revised soil and water erosion equation (RUSLE) (Equation (7)). In this study, the potential soil erosion amount is defined as the supply of soil and water conservation function; the actual soil erosion amount is the demand of soil and water conservation function.

### 2.3.3. Analysis of Key Areas of Ecosystem Services in Shandong Province Based on MSPA

The MSPA model is used to implement the morphological pattern analysis of cultivated land parcels in Shandong Province in 2000 and 2020, in which the cultivated land raster layer is used as the foreground of the MSPA analysis, and the rest of the land classes are used as the background. The operation was carried out under the Guidos Toolbox 2.7 plug-in for ArcGIS 10.2.2. By reclassifying the 2000 and 2020 raster images imported into the software can be obtained. The guidance tutorial of MSPA defines the core area (Core) as the larger habitat patch in the foreground image element, which can provide larger habitats for species, is important for biodiversity conservation, and is an important ecological source area in the ecological network. In this study, the extracted Core areas were considered as ecological source areas, and the top three cultivated plots in terms of the area of ecological source area patches were considered as Core ecological source areas.

### 2.3.4. MCR Model

The minimum cumulative resistance model (MCR model) is a model of the total resistance that a species has to overcome to travel from a 'source' point to a target location. In general, the drag coefficient is defined as a specific value, and the minimum drag value is the source, usually defined as 1. The other factors are determined by considering the actual situation and the objective of establishing the pathway in the study area, and the unit drag coefficient varies according to the objective. The specific formula is expressed as follows:

$$MCR = f_{min} \sum_{j=n}^{i=m} (D_{ij} \times R_i) \tag{12}$$

where, *MCR* is the minimum cumulative resistance of the most indicated source, and *f* denotes the monotonous increasing function to be determined. $D_{ij}$ denotes the spatial distance from landscape unit *i* to *j* in ecological land; $R_i$ is the resistance coefficient of landscape unit *i* to a certain movement; and m and n denote the number of landscape unit i and ecological source *j*, respectively (Table 2).

**Table 2.** Weights of different land use.

| Land-Use | Cultivated Land | Forest | Grassland | Construction | Water | Unused |
|---|---|---|---|---|---|---|
| weights | 3 | 2 | 3 | 5 | 1 | 4 |

The specific steps of calculation are as follows:

- After reclassifying the ESA land use data, determine the resistance values of different land use types according to the following table, with a land use weight of 0.4.
- Using the raster calculator, the total resistance base step file cost is obtained by multiplying the weights of the above diagrams and adding them together.
- Use the cost distance function in ArcGIS to derive the minimum resistance surface data *mcr*. The core ecological zone points identified in the previous section were identified as core ecological nodes, the remaining ecological source zone points became ecological nodes, the cost distance function was used to calculate the total resistance and back links, and the core ecological paths were identified using the cost path function in the ArcGIS.

## 3. Results

### 3.1. Ecosystem Services Supply and Demand

Food production in Shandong Province is greater than food demand from 2000 to 2020, and the margin of safety for food supply is expanding year by year. The difference between annual food production and food consumption grewfrom 6.16 million tons in 2000 to 15.26 million tons in 2020. However, the gap between the food production capacity of each county is obvious. Through the food production supply and demand ratio calculation results shown in the figure, northwestern Shandong's cultivated land food production services in each year supply and demand ratio is higher than the provincial average, thanks to the region's superior food production capacity and lower average population. Priority should be given to protecting the region's food production capacity in the future. The supply-demand ratio for food production is generally low in the central-south Shandong region. In Figure 2, the counties in red on the map are those where local food supply is less than local population demand, and those in green are those where local food supply is greater than local population demand. It can be seen that the counties in Shandong Province show a hollowing out of food self-sufficiency. The supply of food production in northwestern Shandong, southwestern Shandong, and inland areas of the peninsula is greater than the demand. In Central Shandong, due to the taught cultivated land holdings and higher population density, local food production is insufficient to meet its own food needs. The degree of supply and demand of food production is also related to the level of regional economic development. In Jinan and Qingdao, the rapid economic development and the large amount of land used for construction have crowded out cultivated land, resulting in an insufficient supply capacity for food production in the core areas of the two metropolitan areas. In the future, the balance between economic development and food security will be a major issue for the province. There are obvious spatial differences in the match between supply and demand for food production, with a high supply-low demand match in Northwest Shandong, a high supply-high demand match in Southwest Shandong, a low supply-high demand match in Central-South Shandong, and a low supply-high demand match in Peninsula Shandong. In the south-central Shandong province, the match

is high supply–high demand, in the south-central Shandong province, the match is low supply–high demand, and in the peninsula, the match is low supply–low demand.

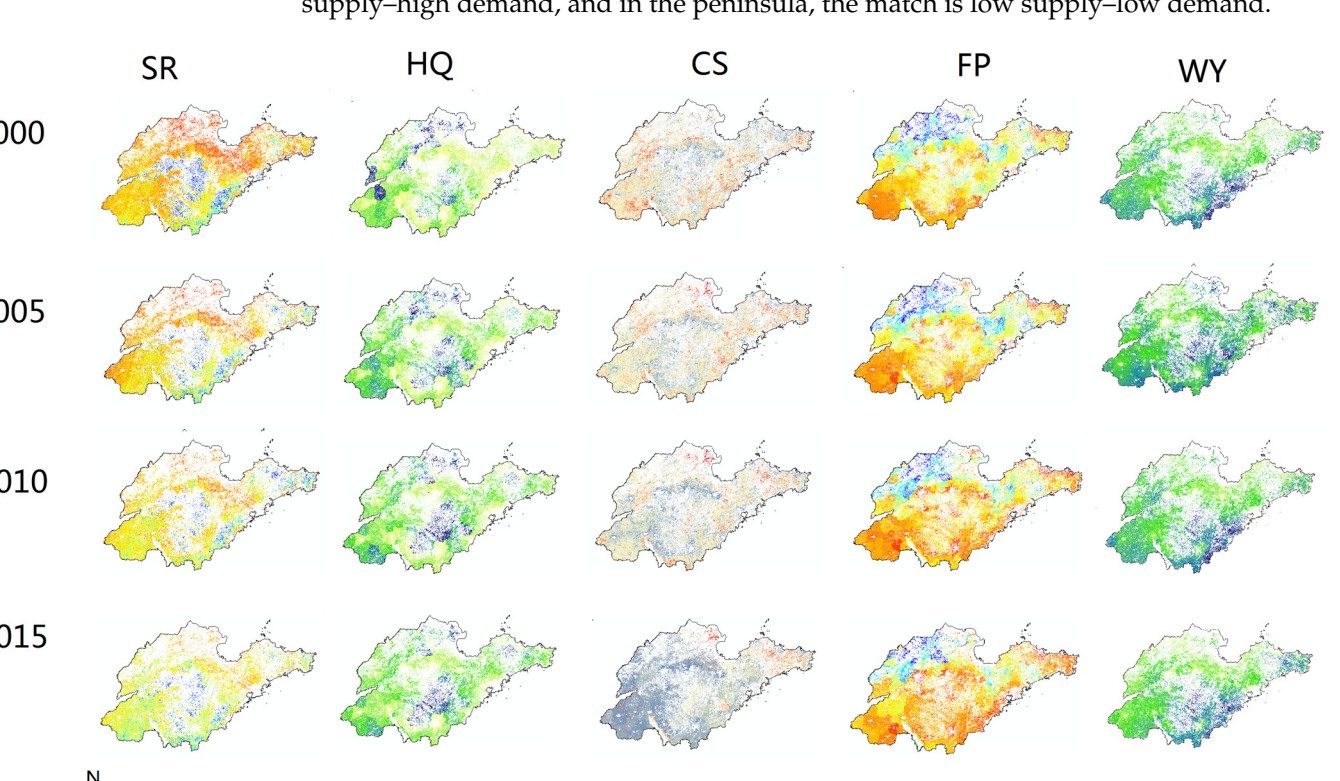

**Figure 2.** Supply-demand ratio of five ESs from 2000 to 2020.

The areas with high water supply and demand in Shandong Province are mainly concentrated in the southeastern coastal region. This distribution is related to higher regional precipitation and lower average population. The regional distribution of the supply-demand ratio of carbon sequestration is similar to the regional distribution of supply, with a high supply-demand ratio in the eastern peninsula and a higher demand for carbon sequestration and oxygen sequestration services than supply in the southwestern part of the country. The spatial pattern of the demand for soil and water conservation in Shandong Province from 2000 to 2020 is not likely to change much. Generally speaking, the central and southern regions of Shandong Province have higher elevations and greater terrain slopes, and the demand for soil and water conservation in this region is higher. The areas with higher supply and demand are distributed in the south and southwestern Shandong regions.

### 3.2. Ecological Corridors

It is calculated that the ecological source area of Shandong Province in 2020 will be 7200 square kilometers less than that of 2000, with a core source area of 6800 km. The number of patches increases by about 120, and the trend of fragmentation of the cultivated land ecological zones is obvious. The percentage of core ecological source area decreases significantly (Figures 3 and 4, Table 3).

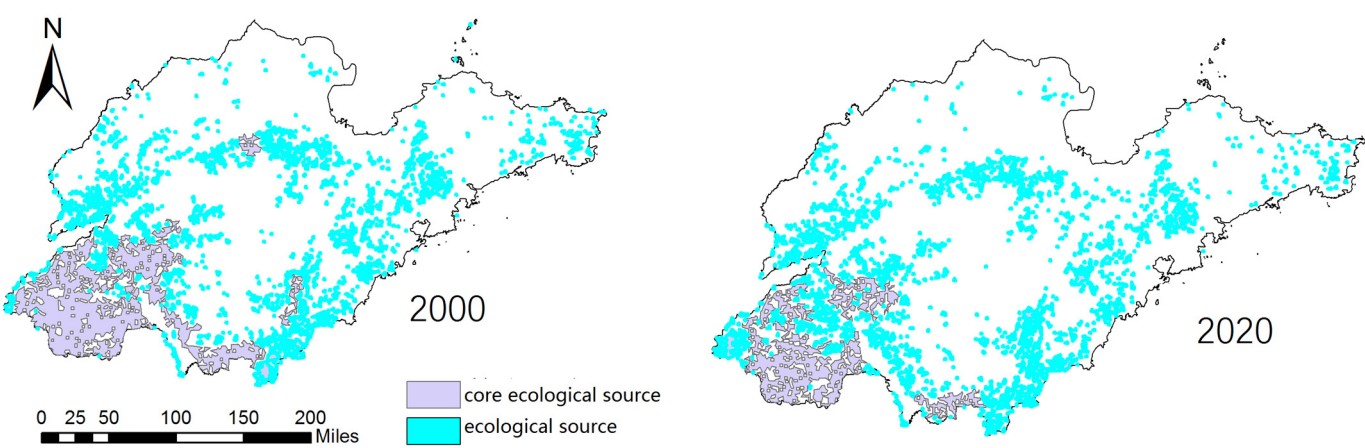

**Figure 3.** Ecological source sites for cultivated land under the MSPA model.

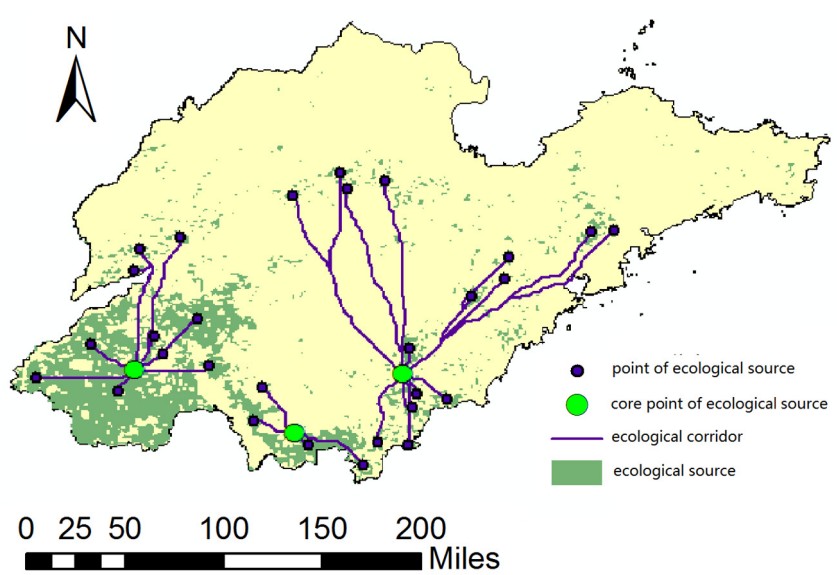

**Figure 4.** Ecological source sites and corridors in Shandong.

**Table 3.** Changes in ecological core parks in Shandong Province, 2000–2020.

| Year | Area of Ecological Source (km$^2$) | Number of Patches | Area of Core Ecological Source (km$^2$) | Percentage |
|------|------|------|------|------|
| 2020 | 14,902.68 | 1057 | 7901.09 | 10.44% |
| 2000 | 22,143.29 | 938 | 14,783.26 | 18.43% |

## 4. Discussion

The spatial and temporal heterogeneity of the cultivated land functions in Shandong Province is obvious, and the imbalance between regional supply and demand ratios is prominent. The ecological protection of the cultivated land is under pressure. The functions of the cultivated land are the various utilities produced by the interrelationship between various cultivated land elements to meet human needs. In this paper, five ecosystem service indicators were selected to evaluate the functionality of the cultivated land system in Shandong Province, including food production, water conservation, carbon sequestration and oxygen release, habitat quality, and soil and water conservation [34]. The results show that the spatial and temporal heterogeneity of cultivated land functions in Shandong Province is obvious. In general, the ecological resource value of cultivated land in the southeast coast of Shandong Province is higher, and the food production function is better in the plain area of northwest Shandong Province. Southwest Shandong has large contiguous cultivated

land plots, which are important ecological sources in terms of landscape morphology. In addition, the imbalance between the supply and demand ratios of cultivated land functional areas in Shandong Province is prominent [7]. Regionally, the southwest Shandong region has a better capacity for carbon sequestration and oxygen release, but a poorer capacity for water conservation. The southeast Shandong Peninsula region has better habitat maintenance capacity and higher water conservation capacity than the provincial average. Changes in the spatial pattern of cultivated land can cause instability in the supply of ecosystem services. In the future, the supply of regional ecosystem services can be coordinated by strengthening the interactions between different ecosystem services to promote the balance between supply and demand and sustainable regional development [35].

The results show that the spatial heterogeneity of the cultivated land functions in Shandong Province is obvious. Generally speaking, the ecological resource value of cultivated land is higher in the southeast coast of Shandong Province, and the food production function is better in the plain area of northwest Shandong Province. Southwest Shandong has a large area of contiguous cultivated land plots, which is an important ecological source from the perspective of landscape morphology. In addition, the imbalance between the supply and demand ratios of cultivated land functional areas in Shandong Province is prominent. Regionally, the southwest Shandong region has a better capacity for carbon sequestration and oxygen release, but a poorer capacity for water conservation. The southeast Shandong Peninsula region has a better habitat maintenance capacity and a higher water conservation capacity than the provincial average. Changes in the spatial pattern of cultivated land can cause instability in the supply of ecosystem services [27]. The total length of ecological corridors is 798.5 km$^2$. The majority of corridors are located in central and southern Shandong

Wu pointed out that the research of landscape sustainable science should focus on the global scale [36]. Only in this way can we better comprehensively analyze the relevant influencing factors, so as to more accurately grasp the spatiotemporal variation laws of ecosystem services. In recent years, the concept of remote coupling has gradually emerged, which also requires researchers to have a broader perspective. Due to the scarcity of relevant data and the limitations of their own research level, this study chose to conduct research at the provincial scale. In the future, based on the existing research, we hope to carry out some national or global scale research, further grasp the spatiotemporal variation laws of the cultivated land system, and use relevant data to analyze and simulate the sustainability of the cultivated land system at the national and global levels.

## 5. Conclusions

In this paper, five ecosystem service indicators were selected to evaluate the multifunction of cultivated land systems in Shandong Province, including food production, water conservation, carbon sequestration and oxygen release, habitat quality, and soil and water conservation. The main conclusions are as follows:

(1) Cultivated land is the main provider of several ecosystem services in Shandong Province. The main source of food production functions is in the southwestern part of the province, water conservation in the southern part of the province, and habitat quality in the central and southern part of the province. In the future, integrated planning and unified regulation can be based on this.

(2) From 2000 to 2020, the fragmentation of cultivated land in Shandong Province is obvious. The area of the ecological source area decreased by nearly 7000 km$^2$. The capacity to provide ecosystem services needs to be further improved. In particular, the protection of the large patches of cultivated land in southwestern Shandong should be strengthened.

(3) The main ecological source areas of cultivated land in Shandong Province are the southwestern and central Shandong mountains and the southern Shandong coast. In practice, it is necessary to strengthen the interconnection between the source areas and protect the key ecological corridors.

Although this study provides guidance for food production areas in terms of ecological protection of cultivated land, it lacks a multi-factor co-analysis in the construction of the resistance face. In the future, further improvements will be made in this area. Meanwhile, the supply of regional ecosystem services can be coordinated by strengthening the interactions between different ecosystem services to promote the balance between supply and demand and the sustainable development of the region. In general, this study will provide academic grounds for policy makers and authorities on this topic.

**Author Contributions:** Conceptualization, W.Q.; methodology, Y.X. and Y.L.; software, Y.L.; validation, Y.X. and Y.L.; formal analysis, W.Q.; investigation, Y.L. and Q.S.; resources, W.Q.; data curation, Y.L.; writing—original draft preparation, Y.L.; writing—review and editing, Y.L.; visualization, W.Q.; supervision, W.Q. All authors have read and agreed to the published version of the manuscript.

**Funding:** This research was funded by the remote sensing monitoring of cultivated land and establishment of its benchmark land prices, grant number 381180/010, and the open fund of the Key Laboratory of Land Surface Pattern and Simulation, the Chinese Academy of Sciences, grant number LBKF201802, and the Social Science Planning Project of Shandong Province, grant number 22CGLJ40.

**Institutional Review Board Statement:** Not applicable.

**Informed Consent Statement:** Not applicable.

**Data Availability Statement:** The raw data supporting the conclusions of this article will be made available by the authors on request.

**Acknowledgments:** The authors would like to thank the editors and reviewers for their helpful comments.

**Conflicts of Interest:** The authors declare no conflict of interest.

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
