# Peer review of "Construction of Cultivated Land Ecological Network Based on Supply and Demand of Ecosystem Services and MCR Model: A Case Study of Shandong Province, China"

_sustainability, doi:10.3390/su16093745_

Round 1

Reviewer 1 Report

Comments and Suggestions for Authors

In the submitted paper, the ecosystem service capacity of the cultivated land system in Shandong Province was analyzed and evaluated by calculating the supply and demand of ecosystem services such as food production, water conservation, carbon sequestration and oxygen release, habitat quality, soil, and water conservation. Then, a spatial presentation of the demand-supply relationship of the analyzed services in Shandong Province was carried out and the corresponding spatial-temporal changes from 2000 to 2020 were identified. Using the MSPA model, the core area of the farmland ecosystem was divided based on morphological analysis. Based on the results of this research, the MCR model was developed to analyze and calculate the ecological security pattern of farmland in Shandong Province. The research results indicate the areas and directions of ecological protection of cultivated land systems in the study province of China. Taking into account the needs of the international reader, it is suggested that the following additions be made to the original text:

1) In the Introduction, please explain in more detail the key concepts in Chinese reality such as quantity and quality protection of cultivated land and ecological protection of cultivated land, or ecosystem services of cultivated land;

2) In addition, the aim of the research, as well as the research gap that the conducted research aims to fill, should be made more explicit in the Introduction. Furthermore, the relationship between the research carried out and the concept of sustainability should be indicated.

3) In the Conclusions, several general (utilitarian) conclusions should be formulated, which would bring new values to all researchers dealing with the protection of cultivated land in the world. These may be conclusions of both methodological and substantive nature. 

Once these additions have been made and the text has been adapted to the editorial requirements, I recommend the paper for publication in the journal Sustainability.

Author Response

Response to reviewer 1

Comments1: “In the Introduction, please explain in more detail the key concepts in Chinese reality such as quantity and quality protection of cultivated land and ecological protection of cultivated land, or ecosystem services of cultivated land;”

Response: thanks for your invaluable advice, the Introduction part has been revised .

Comments 2:“In addition, the aim of the research, as well as the research gap that the conducted research aims to fill, should be made more explicit in the Introduction. Furthermore, the relationship between the research carried out and the concept of sustainability should be indicated.”

Response: thanks for your invaluable advice, the introduction part has been revised.

Comments 3:“In the Conclusions, several general (utilitarian) conclusions should be formulated, which would bring new values to all researchers dealing with the protection of cultivated land in the world. These may be conclusions of both methodological and substantive nature.”

Response: thanks for your invaluable advice, the Conclusions part has been rewritten.

Reviewer 2 Report

Comments and Suggestions for Authors

Paper title: Construction of Cultivated Land Ecological Network Based on Supply and Demand of Ecosystem Services and MCR Model: A Case Study of Shandong Province, China

The observations are:

1) Line 146. 

How the NDVI response is equivalent to Food production (FP)?

If this hypothesis is true, the green cover means growing areas.

2) Line 194. 

The equation 3 (WY = P −ET) is a water balance. 

The authors must explain this argument based only on water surface disponibility.

3) Figure 2. Supply and demand ratio of five ESs from 2000 to 2020.

The figure 2 has an unsatisfying resolution. It is impossible to analyze.

4) Methods.

2.3.1. The ecosystem service supply indicators: all the equations must have a reference author. Some of them don´t have.

5) Discussion.

The authors do not discuss the results with the literature.

The paper has an approach relevant but needs adjustments, especially in methodology, results, and discussion.

Maps should be enhanced for better resolution.

Author Response

Response to reviewer 2

Comments1: “ Line 146. How the NDVI response is equivalent to Food production (FP)?If this hypothesis is true, the green cover means growing areas.”

Response: thanks for your invaluable advice, the correlation between NDVI and food production  has been tested by other scholars, a new reference has been added.

Comments 2: “ Line 194. The equation 3 (WY = P −ET) is a water balance. The authors must explain this argument based only on water surface disponibility.”

Response: thanks for your invaluable advice, this part has been revised.

Comments 3:“Figure 2. Supply and demand ratio of five ESs from 2000 to 2020.The figure 2 has an unsatisfying resolution. It is impossible to analyze.”

Response: thanks for your invaluable advice,All pictures have been modified.

Comments 4: “Methods.2.3.1. The ecosystem service supply indicators: all the equations must have a reference author. Some of them don´t have.”

Response: thanks for your invaluable advice, new references have been added

Comments 5: “Discussion.The authors do not discuss the results with the literature. The paper has an approach relevant but needs adjustments, especially in methodology, results, and discussion. Maps should be enhanced for better resolution.”

Response: thanks for your invaluable advice, the Discussion part has been rewritten. All pictures have been modified.

Reviewer 3 Report

Comments and Suggestions for Authors

The study used several models and methods to carry out ecological network construction of arable land based on ecosystem services. The starting point of the study is relatively novel, but there are some problems in the experimental design and description, which have not yet reached the standard required by the journal. The main issues are as follows:

1. The background and significance of the abstract are not well in-depth and lack the need for the study. In addition, the abstract focuses on a discussion of the research methodology and does not delve into the findings, which is highly inappropriate.

2. In the introduction, paragraphs are disjointed from paragraph to paragraph, and there is no progression of a relationship. It is suggested that the method should be combined with previous studies when discussing the research background, and the present content is too simple. The following references are recommended:

Ecological risk assessment and prediction based on scale optimization- A Case Study of Nanning, a landscape garden city in China

The Spatiotemporal Variation in Biodiversity and Its Response to Different Future Development Scenarios: A Case Study of Guilin as an Internationally Renowned Tourist Destination in China

Exploring the Drivers of Soil Conservation Variation in the Source of Yellow River under Diverse Development Scenarios from a Geospatial Perspective

3. The last paragraph of the introduction suggests expanding the discussion by points.

4. The data used for the study is more, and it is recommended that a table description be added in subsection 2.2, which will provide information on the resolution of the data that needs to be added.

5. Figure 1 needs to be recreated, it lacks cartographic elements such as latitude and longitude grids, and the map of China on the left appears to be a screenshot, which is very uncritical.

6. Suggest adding a technology roadmap to clarify the overall research framework.

7. In the fifth part (SR) of subsection 2.3.1, why was the P factor not considered in the construction of the soil loss model? If the effect of this factor is ignored in this study, then this must be emphasized in the original text.

8. There are some problem in the construction of the resistance surface, it is unreasonable to use only land use as the resistance value, we should consider the status quo of the study area comprehensively, and choose multiple factors to construct a comprehensive resistance value.

9. Figure 2 elements are blurred, please readjust, why are there two scales? Also, why is there such a big difference in the spatial distribution between 2015 and 2020 in the CS analysis? Have the authors thought seriously about this.

10. In subsection 3.2, the authors do not seem to have analyzed the distribution of ecological corridors in the study area and the ecological network analysis is incomplete.

11. The conclusion section should be cohesive and provide important insights into the current field, and suggested points to develop the argument.

Author Response

Response to reviewer 3

Comments 1: “ The background and significance of the abstract are not well in-depth and lack the need for the study. In addition, the abstract focuses on a discussion of the research methodology and does not delve into the findings, which is highly inappropriate.”

Response: Thanks for your invaluable advice, the Introduction part has been revised .

Comments 2: “In the introduction, paragraphs are disjointed from paragraph to paragraph, and there is no progression of a relationship. It is suggested that the method should be combined with previous studies when discussing the research background, and the present content is too simple. The following references are recommended.”

Response: Thanks for your invaluable advice, the introduction part has been revised.

Comments 3: “The last paragraph of the introduction suggests expanding the discussion by points.”

Response: Thanks for your invaluable advice,  the Introduction part has been revised.

Comments 4: “The data used for the study is more, and it is recommended that a table description be added in subsection 2.2, which will provide information on the resolution of the data that needs to be added. The last paragraph of the introduction suggests expanding the discussion by points.”

Response: Thanks for your invaluable advice,  data usage part has been revised.

Comments 5: “Figure 1 needs to be recreated, it lacks cartographic elements such as latitude and longitude grids, and the map of China on the left appears to be a screenshot, which is very uncritical.”

Response: Thanks for your invaluable advice,  the picture has been revised.

Comments 6: “Suggest adding a technology roadmap to clarify the overall research framework.”

Response: Thanks for your invaluable advice,  the roadmap has been added.

Comments 7. “In the fifth part (SR) of subsection 2.3.1, why was the P factor not considered in the construction of the soil loss model? If the effect of this factor is ignored in this study, then this must be emphasized in the original text.”

Response: “Thanks for your invaluable advice, considering this research focus on the soil retention function of cultivated land only, the P factor is 0.5. the equation(7) has been modified

Comments 8. “There are some problem in the construction of the resistance surface, it is unreasonable to use only land use as the resistance value, we should consider the status quo of the study area comprehensively, and choose multiple factors to construct a comprehensive resistance value.”

Response: Thanks for your invaluable advice. unfortunately, due to the limitation of time and resources, we are unable to reconstruct resistance surface by utilizing multiple factors. We do plan to apply your advice to improve our research in the future .

Comments 9.“Figure 2 elements are blurred, please readjust, why are there two scales? Also, why is there such a big difference in the spatial distribution between 2015 and 2020 in the CS analysis? Have the authors thought seriously about this.”

Response: Thanks for your invaluable advice,  the picture has been modified.

Comments 10. “In subsection 3.2, the authors do not seem to have analyzed the distribution of ecological corridors in the study area and the ecological network analysis is incomplete.”

Response: Thanks for your invaluable advice,  the Analysis part has been rewritten.

Comments 11. “The conclusion section should be cohesive and provide important insights into the current field, and suggested points to develop the argument.”

Response: Thanks for your invaluable advice,  the Conclusion part has been rewritten.

Round 2

Reviewer 2 Report

Comments and Suggestions for Authors

Dear,

Figures in the assessment text are still in low resolution. I think the resolution will be adjusted in the final version. The discussion remains with the author's evaluation of the results without comparisons with the literature.

Author Response

thanks for your comments, changes have been made

Reviewer 3 Report

Comments and Suggestions for Authors

Few changes were made to the new manuscript, and the questions raised in the first round were almost ignored.

Author Response

Sincerely apologies to reviewer3, wrong file was uploaded in previous round

Round 3

Reviewer 3 Report

Comments and Suggestions for Authors

The quality of the paper has been significantly improved after modification according to the opinions. It is suggested to quote the following latest literature in the introduction:

Ling, M.; Chen, J.; Lan, Y.; Chen, Z.; You, H.; Han, X.; Zhou, G. Exploring the Drivers of Soil Conservation Variation in the Source of Yellow River under Diverse Development Scenarios from a Geospatial Perspective. Sustainability 2024, 16, 777. https://doi.org/10.3390/su16020777

Lan, Y.; Zhang, K.; Han, X.; Chen, Z.; Ling, M.; You, H.; Chen, J. The Spatiotemporal Variation in Biodiversity and Its Response to Different Future Development Scenarios: A Case Study of Guilin as an Internationally Renowned Tourist Destination in China. Appl. Sci. 2024, 14, 2101. https://doi.org/10.3390/app14052101

Chen, J.; Yang, Y.; Feng, Z.; Huang, R.; Zhou, G.; You, H.; Han, X. Ecological Risk Assessment and Prediction Based on Scale Optimization—A Case Study of Nanning, a Landscape Garden City in China. Remote Sens. 2023, 15, 1304. https://doi.org/10.3390/rs15051304